# Acquisition of the Midbrain Dopaminergic Neuronal Identity

**DOI:** 10.3390/ijms21134638

**Published:** 2020-06-30

**Authors:** Simone Mesman, Marten P. Smidt

**Affiliations:** Swammerdam Institute for Life Sciences, FNWI, University of Amsterdam, 1098 XH Amsterdam, The Netherlands; s.mesman@uva.nl

**Keywords:** mdDA neuronal development, floor plate patterning, DA progenitor, neuronal differentiation, subset specification

## Abstract

The mesodiencephalic dopaminergic (mdDA) group of neurons comprises molecularly distinct subgroups, of which the substantia nigra (SN) and ventral tegmental area (VTA) are the best known, due to the selective degeneration of the SN during Parkinson’s disease. However, although significant research has been conducted on the molecular build-up of these subsets, much is still unknown about how these subsets develop and which factors are involved in this process. In this review, we aim to describe the life of an mdDA neuron, from specification in the floor plate to differentiation into the different subsets. All mdDA neurons are born in the mesodiencephalic floor plate under the influence of both SHH-signaling, important for floor plate patterning, and WNT-signaling, involved in establishing the progenitor pool and the start of the specification of mdDA neurons. Furthermore, transcription factors, like Ngn2, Ascl1, Lmx1a, and En1, and epigenetic factors, like Ezh2, are important in the correct specification of dopamine (DA) progenitors. Later during development, mdDA neurons are further subdivided into different molecular subsets by, amongst others, Otx2, involved in the specification of subsets in the VTA, and En1, Pitx3, Lmx1a, and WNT-signaling, involved in the specification of subsets in the SN. Interestingly, factors involved in early specification in the floor plate can serve a dual function and can also be involved in subset specification. Besides the mdDA group of neurons, other systems in the embryo contain different subsets, like the immune system. Interestingly, many factors involved in the development of mdDA neurons are similarly involved in immune system development and vice versa. This indicates that similar mechanisms are used in the development of these systems, and that knowledge about the development of the immune system may hold clues for the factors involved in the development of mdDA neurons, which may be used in culture protocols for cell replacement therapies.

## 1. Introduction

The mesodiencephalic dopaminergic (mdDA) group of neurons is involved in motor control, motivation, reward, and addiction. In patients with Parkinson’s disease (PD), a specific part of the mdDA neuronal population, the substantia nigra (SN), degenerates, resulting in motor (e.g., bradykinesia and rest tremor) and non-motor (e.g., cognitive/neuro-behavioral disorders and sensory and sleep abnormalities) features [1,2]. Interestingly, although the SN degenerates during PD, the ventral tegmental area (VTA) remains relatively intact, indicating that the mdDA neuronal population consists of different subsets of neurons, each with a distinct vulnerability for neuronal degeneration [1,3,4]. Significant research has been conducted on the development of mdDA neurons and the distribution into the different subsets, as the development of mdDA neurons for transplantation in the treatment of PD is dependent on the generation of the correct subset of neurons [4,5,6].

MdDA neurons are defined by the expression of general markers, like tyrosine hydroxylase (TH), the rate-limiting enzyme in dopamine (DA) production, vesicular monoamine transporter 2 (VMAT2), and paired-like homeodomain 3 (PITX3), and subset-specific markers, like dopamine transporter (DAT), aldehyde dehydrogenase 2 (AHD2), and pre-B-cell leukemia homeobox 3 (PBX3) for neurons of the SN, and cholecystokinin (CCK) and WW domain-containing oxidoreductase (WWOX) as markers for neurons of the VTA [3,4,7,8]. In this review, we aim to describe the life of an mdDA neuron in the murine brain (unless otherwise specified), from specification in the floor plate (FP) to differentiation into the various subsets in the mdDA group of neurons. Furthermore, we will try to link the development of the mdDA neuronal population to immune system development, as both systems comprise different subsets of cells, which are derived from floor plate (FP)-derived progenitors.

## 2. Patterning of the Midbrain Area

### 2.1. Specification of the Isthmic Organizer

MdDA neurons are thought to arise from the FP and FP-basal plate (BP) boundary from E10.5 onward in the embryonic midbrain [9,10,11,12], whereas the hindbrain is the place of origin of serotonergic neurons [13]. This indicates that the mid- and hindbrain are two separate structures that express different intrinsic and extrinsic signals during development. The distinction between these structures is dependent on the correct development of the mid–hindbrain border, also known as the isthmus [14,15]. The isthmus develops around E7.5 at the intersection of *Otx2* expression in the midbrain and *Gbx2* expression in the hindbrain (Figure 1A) [14,16,17,18]. The sharp border between the expression areas of these two genes is established by the dose-dependent repressional action of *Otx2* on *Gbx2* and vice versa. Loss of *Otx2* results in an extension of *Gbx2* into midbrain areas, which consequently obtain a hindbrain-specific fate, whereas loss of *Gbx2* results in an extension of *Otx2* into the hindbrain, resulting in an expansion of the midbrain into the hindbrain, indicating that these factors are of major importance for the correct specification of these brain structures [14,15,18]. The isthmus itself is defined by expression of *Fgf8* within the isthmus and expression of *Wnt1*, which protrudes into the midbrain and is importantly involved in the development of mdDA neurons (Figure 1A) [17,19,20,21,22,23,24]. Interestingly, although the bordered expression of *Fgf8* in the isthmus is dependent on the expression areas of *Otx2* and *Gbx2*, expression of *Fgf8* itself is not regulated by either one [21,25], indicating that this is controlled by other factors during isthmus specification. One of the factors that is thought to regulate the expression of *Fgf8* is Engrailed-1 (*En1*) (Figure 1A). It has been shown that ectopic expression of *En1* in the midbrain and diencephalon can induce the expression of *Fgf8*, and loss of *En1* has an effect on the differentiation of neuronal populations in both the mid- and hindbrain [26,27,28]. Furthermore, it was recently shown that *En1* is essential in regulation and maintenance of the proper expression of *Fgf8* and other isthmus-related genes like *Wnt1* and *Otx2* [29]. Together, these data point to an essential role of *En1* in early embryonic development and correct establishment of the isthmus.

Besides a direct role for transcription factors in the establishment of the isthmus, evidence of epigenetic regulation of isthmus patterning is increasing. One epigenetic factor recently shown to influence isthmus patterning is Enhancer of zeste homolog 2 (*Ezh2*)*,* which is part of the Polycomb repressional complex and regulates gene expression by trimethylation of lysine 27 of histone 3 (H3K27Me3), an overall repressive mark [30]. Conditional deletion of *Ezh2* in the *En1* expression area results in a disorganized isthmus, characterized by ectopic expression of *Otx2* in the hindbrain and a loss of *Fgf8* and *Wnt1* in the isthmic area [31]. The phenotypic consequences of the loss of *Ezh2* resemble the effects on the isthmus as described for the *En1* mutant [29]. However, it is likely that these genes function via different pathways, as *En1* expression was not affected in the *Ezh2* mutant [31]. These data indicate that, besides the known transcription factors involved in isthmus patterning, other molecular cascades, e.g., epigenetic regulation, are importantly involved in isthmus development, which may function independently of currently known factors.

### 2.2. FP Patterning in the Midbrain

After establishment of the isthmus, the FP is specified in the developing midbrain. In the embryonic midbrain, the FP is likely induced via GLI2A-mediated SHH signaling, as *Gli2* mutant mice do not develop a FP in the midbrain, hindbrain, and spinal cord [32]. MdDA neurons are thought to arise in the FP and FP-BP boundary of the midbrain at the intersection of FGF8 secreted by the isthmus and SHH secreted by the FP [33,34,35]. After neuronal birth, mdDA neurons migrate from the ventricular zone (VZ) of the FP toward their final position in the mesodiencephalon [9,11,12,36,37]. SHH-signaling is mainly necessary for the correct patterning of the FP, whereas WNT-signaling is involved in the induction of proliferation via inhibition of *Shh* expression, resulting in a large progenitor pool from which mdDA neurons originate (Figure 1B) [12,38,39,40,41,42]. Further underlining the importance of SHH-signaling in FP patterning and the consequent correct development of mdDA neurons is the fact that loss of SHH-signaling from E9.5 onward does not have a gross effect on the development of mdDA neurons [41], which are born from E10.5 onward [9,43], whereas loss of *Shh* from E7.5 onward in the developing FP dramatically reduces the amount of differentiated mdDA neurons [41,44]. This difference in the effect of *Shh*-deletion on mdDA neuronal development is likely due to the fact that early (~E7.5) loss of *Shh* leads to a smaller or different neural progenitor pool and, consequently, a decrease in the amount of mature mdDA neurons, whereas at later stages (~E9.5), the FP patterning is complete, the DA progenitor pool is completely developed, and loss of *Shh* only has a small, if any, effect on the size of the progenitor pool, resulting in a relatively normal development of mdDA neurons (Figure 1B). Furthermore, it has been suggested that SHH inhibits FP neurogenesis and that WNT-signaling inhibits SHH in the FP to induce FP neurogenesis, DA progenitor expansion, and the expression of DA progenitor-specific genes, like *Lmx1a* (Figure 1C) [10,19,38,45]. This reciprocal effect of SHH and WNT is nicely illustrated in studies performed in the hindbrain, where a DA cell-fate is repressed by the continuous expression of *Shh* in the FP. Overexpression of *Wnt1* in this brain structure results in an inhibition of *Shh* expression and initiation of mdDA neurogenesis [22,46].

Besides *Shh* and *Wnt1,* other genes expressed in the midbrain FP are importantly involved in the onset of mdDA neuronal specification, like the LIM-homeodomain transcription factor *Lmx1a* and the basic helix-loop-helic (bHLH) factors *Ngn2* and *Ascl1* (*Mash1*) [10]. In chick, *Lmx1a* expression in the mesodiencephalic FP upon the start of mdDA neuronal differentiation is induced by SHH-signaling and shown to be of crucial importance in the development of the mdDA neuronal population [47,48]. This is unlike what was detected in chick, in the mouse embryo not SHH-signaling, but canonical WNT-signaling may be essential for the correct expression of *Lmx1a* in the mesodiencephalic FP, as WNT-signaling regulates expression of *Oc1*, a regulator of *Lmx1a* expression, and ectopic expression of β-catenin in the FP results in an upregulation of *Lmx1a* in rostral regions (Figure 1C) [45,49,50]. Interestingly, ectopic expression of β-catenin and, consequently, the rostral extension of *Lmx1a* in the developing FP did not result in an increase in TH^+^ neurons but rather had an adverse effect [45], indicative of the necessity of correct FP patterning, timing of gene expression, and regulation of WNT-dosage for the induction of mdDA specification of neural progenitors in the midbrain. Furthermore, deletion of *Lmx1a* in the murine midbrain does not result in a complete loss of mdDA neurons, but only shows a reduction in the amount of TH^+^ mdDA neurons [48,51,52,53], suggesting a role for *Lmx1a* in efficient proliferation and/or neurogenesis of the mdDA progenitor pool in the murine developing embryo. The fact that not all TH^+^ mdDA neurons are lost in the *Lmx1a* murine mutant, as was detected in the chick *Lmx1a* mutant, may be attributed to compensation by *Lmx1b* in the murine midbrain [51,53].

*Lmx1a* may define its role in proliferation and neurogenesis through regulation of the expression of the bHLH gene *Ngn2*, as overexpression of *Lmx1a* leads to an increase in *Ngn2* expression both in vitro and in vivo, although it is not clear whether this is a direct or indirect effect [45,54]. *Ngn2*, together with *Ascl1*, is known to play a role in patterning of the FP and, later, the start of neurogenesis. In the developing midbrain, *Ascl1* and *Ngn2* are found to act together in the correct development of mdDA neurons [55]. *Ascl1* is thought to be involved in the proliferation and gain of the neurogenic potential of neural precursors in the FP [56] (Figure 1C). *Ascl1* mutants show a reduction in neurogenesis in the FP area, and overexpression of this factor in vitro leads to increased neurogenic potential of neural progenitor cells (NPCs) without depletion of the naïve NPC pool [56]. *Ngn2*, on the other hand, was shown to induce rapid neurogenesis in vitro upon overexpression, which involved one terminal cell division, thereby depleting the NPC pool (Figure 1C) [56,57]. bHLH proteins, in general, have been shown to be important in neuronal differentiation and proliferation throughout the embryo and specifically in the brain [58,59,60,61]. Recently, other bHLH proteins have been identified to be involved in FP patterning, expansion of the progenitor pool, and mdDA neurogenesis during embryogenesis [62,63,64,65]. The bHLH protein *Nato3* is expressed in the FP throughout development and has been shown to be involved in the regulation of multiple genes involved in mdDA neurogenesis and mdDA neuron lineage commitment [62,64]. Loss of *Nato3* in the FP leads to a decrease in mdDA neurons, suggesting that it could be importantly involved in expansion of the progenitor pool in the murine FP [62]. Interestingly, the bHLH protein *Tcf12* is strongly expressed in the FP and in DA progenitors but does not affect proliferation in the midbrain FP [63]. However, it does affect the expression of early mdDA differentiation genes, like *Lmx1a* and *Nurr1*, and *En1-Cre*-driven deletion of this gene results in a delay in mdDA neuronal differentiation, indicating that this gene is involved in the correct specification of DA progenitors [63]. Taken together, these studies point to a complex regulation of FP patterning and DA progenitor specification in which timing, function, and expression of bHLH factors are crucially involved. As the bHLH protein family is a large superfamily containing many different transcription factors, it would be interesting to further investigate the role of this family in mdDA neuronal development.

## 3. The Road to Adulthood: From NPC to Mature MdDA Neuron

### 3.1. Early Specification of DA Progenitors

As discussed above, correct specification of the midbrain FP is necessary to allow for a large NPC pool that will form the base of DA progenitors and the future mdDA neuronal population. Many factors that are involved in the patterning of the midbrain area have also been linked to the generation of mdDA neurons, indicating that these factors serve a dual function during mdDA neurogenesis, both in the correct patterning and expansion of the NPC pool, and in mdDA cell-fate commitment. One of the earliest patterning factors in the midbrain, *Otx2*, has been found to play such a dual role. Targeted deletion of *Otx2* from E7.5 onward via an *En1-Cre* driver results in a depletion of the mdDA neuronal pool, mainly in the caudal region [66,67]. It is suggested that *Otx2* exerts its function in mdDA neuronal specification by controlling the identity of neural progenitors toward a DA progenitor fate via repression of *Nkx2.2* in the ventral midbrain and maintenance of *Nkx6.1* expression by inhibition of *Shh* [67,68]. Interestingly, if *Otx2* is deleted at E10.5 via a *Nestin*-*Cre* driver, *Shh* expression is not affected, although mdDA neurons are severely decreased in number, indicating that *Otx2* is necessary to obtain the correct amount of mdDA neurons [69]. *Otx2* might positively control the number of mdDA neurons later during embryogenesis via regulation of neurogenesis, by controlling proneural gene expression in and proliferation of DA progenitors [69]. This function could be exerted through regulation of *Ngn2* expression, which would affect the pool of NPCs in the ventral midbrain (Figure 2A) [56,70,71].

An early factor important in the generation of mature mdDA neurons from post-mitotic DA precursors is *Nurr1*. *Nurr1* is crucial in regulation of the expression of *Th*, *Vmat2*, and *Dat*, and loss of *Nurr1* results in a depletion of the mdDA neuronal population [72,73]. It is one of the first factors that is not expressed in the VZ of the FP in the embryos, but in neurons that acquired a DA cell-fate [72,73]. Expression of *Nurr1* has been suggested to be promoted by *Ngn2*, as *Ngn2* is thought to be required for the generation of post-mitotic NURR1^+^ precursors from SOX2^+^ progenitors in the FP (Figure 2A) [55]. Although loss of *Ngn2* affects the amount of TH-expressing neurons in the midbrain, it does not deplete the entire NURR1^+^ pool, indicating that other factors, like *Ascl1* or *Lmx1a*, may also be important in establishing post-mitotic NURR1-expressing DA precursors [51,55,70]. *Ngn2* has been shown to be expressed in a small subset of the *Nurr1*-expressing DA precursors [55], and studies have shown that expression of both *Ngn2* and *Nurr1* increased the probability of mdDA neuronal generation in vitro [56,74,75]. Interestingly, overexpression of *Nurr1* results in immature neurons that express TH but do not have the characteristics of fully mature mdDA neurons, whereas co-expression with *Ngn2* results in the development of mature mdDA neurons [56]. These studies indicate that intermediate DA progenitors shortly express both *Nurr1* and *Ngn2*, which further differentiate into fully matured mdDA neurons after downregulation of *Ngn2*. These data show that mdDA neuronal specification factors work in concert to generate a fully matured mdDA neuron, and in order to accomplish this in vitro, it is necessary to get a proper insight into the actions and spatio-temporal expression of such specification factors.

Besides cell-intrinsic factors that prime NPCs to progress into the DA progenitor state and ultimately to an mdDA neuronal cell-fate, cell-extrinsic factors, e.g., SHH- and WNT-signaling, in the embryonic midbrain are thought to be involved in the correct specification of these neurons [10,19]. As SHH-signaling is mainly important for initiating the progenitor pool at early developmental stages, and mdDA neuronal differentiation appears to be initiated by WNT-signaling, although this role of WNT signaling has not been completely established yet [12,38,76], we will focus on the role of WNT-signaling, both canonical and non-canonical, in mdDA neuronal development.

As discussed above, the earliest expression of a member of the WNT family, *Wnt1*, in the midbrain is detected in the isthmus, from which *Wnt1*-expression is protruding into the midbrain [21,22,23,35]. It is thought that *Wnt1* inhibits the expression of *Shh*, thereby inducing NPCs to acquire a DA progenitor cell-fate [38,46]. Underlining the necessity of WNT-signaling and the possibility of an inhibitory effect of WNT1 on SHH-signaling in order to activate DA specification is the fact that newly formed mdDA neurons show no evidence of SHH-signaling in the Gli1-GFP reporter model for Gli2A-mediated SHH-signaling, whereas the BATGAL reporter model for canonical WNT-signaling showed that these newly formed neurons did receive canonical WNT-signaling [12].

Many different *Wnt*-family members are expressed in the embryonic midbrain, which exert unique and overlapping functions in mdDA neuronal development. Both canonical and non-canonical WNT-signaling have been linked to the development of mdDA neurons, although they are thought to be important at different stages of mdDA neurogenesis. *Wnt1* is linked to the onset of mdDA neurogenesis and correct patterning of the midbrain FP (Figure 1A and Figure 2A). *Wnt1* has been shown to regulate the expression of *En1*, and *Wnt1*-mutants display a similar phenotype as *En1*-mutants [77]. WNT1, WNT2, and WNT3A activate the canonical WNT-signaling pathway via interaction with LRP5 and LRP6, which is involved in the proliferation and differentiation of postmitotic DA precursors into mdDA neurons [78,79,80,81,82]. Deletion of the WNT-receptor LRP6 shows an initial decrease in the amount of TH-expressing mdDA neurons, although this effect recovers over time, suggesting that LRP6 function may be compensated by LRP5 or other WNT-effectors that are expressed in the embryonic midbrain [80,83]. During expansion of the progenitor pool in the FP, the effects of *Wnt1* signaling are counteracted by the expression of FP-specific miRNAs (e.g., miR135a2 and miR34b/c) [84,85]. These miRNAs are shown to downregulate levels of *Wnt1* in DA progenitors, thereby allowing for differentiation of the DA progenitors to mdDA neurons [84,85]. Besides the regulation of *Wnt1* transcript levels in DA progenitors, it has been shown that the canonical WNT-signaling inhibitor DKK1 is involved in mdDA differentiation as the loss of this gene results in a loss of mdDA neurons and affects the distribution of mdDA neurons in the midbrain from E13.5 onward [86]. These data suggest that time-dependent modulation and fine-tuning of the canonical WNT-signaling pathway is important in mdDA neuronal development, and that the balance between the regulation of neurogenesis and neuronal differentiation is crucial in determining the amount of mdDA neurons and subset differentiation.

WNT5A activates the non-canonical WNT-signaling pathway in the midbrain via interference of LRP5 and LRP6 activation upon canonical WNT-signaling, thereby favoring non-canonical WNT-signaling [78]. It has been found that in neural cultures of mouse embryonic stem cells, inhibition of the canonical WNT-signaling enhances neuronal and mdDA differentiation, and loss of the non-canonical *Wnt5a* results in an increase in proliferation of NPCs and a changed distribution of mdDA neurons [87,88]. Loss of *Wnt5a* delays the differentiation of NURR1^+^ precursors into fully matured mdDA neurons, suggesting a role for non-canonical WNT-signaling in terminal differentiation of these neurons [87]. It is likely that different WNT-signaling cascades balance or compensate for each other, as the *Wnt5a*-*Lrp6* double mutant shows a severe phenotype in which proliferation and differentiation in the neural tube are heavily affected [89]. Taken together, these studies show that the exact role of WNT-signaling during mdDA neuronal specification is still unclear. However, it is likely that the effect of canonical and non-canonical WNT-signaling on mdDA neuronal specification is strongly dependent on the time and place during embryogenesis and the sensitivity of DA precursors for WNT-signaling.

### 3.2. Late Differentiation and MdDA Subset Specification

Above, we have provided a short overview of the main factors in the patterning of the midbrain and specification of the DA progenitors. Based on literature, it is clear that most mdDA neurons arise from the caudal FP and FP–BP border in the developing midbrain [9,12,43]. However, the mdDA neuronal population is not a uniform population but consists of different subsets, which express a variety of specific markers and are dependent on a unique set of transcription factors and signaling cascades for their specification [7,8,90,91]. Therefore, an important question is how DA progenitors, which are specified by a relatively small set of factors, acquire a subset-specific fate. A crucial factor in development of the different mdDA neuronal subsets is the time of birth. It was shown that neurons encompassing the SNc are born before neurons that are part of the VTA, indicating that early during DA neurogenesis, DA progenitors acquire a SN cell-fate, whereas late during neurogenesis, VTA-specific neurons are generated [9,10,37,43]. Interestingly, mutants that have a delay in mdDA neuronal differentiation, like the *Ngn2*, *Ascl1*, *Tcf12*, and *Wnt5a* mutants, show a specific depletion of mdDA neurons in the SN, indicating that time of birth has an effect on the migration and differentiation of mdDA neurons [47,55,63,87].

Many factors that are involved in the enlargement of the NPC pool and patterning of the midbrain FP have been found to play a role in mdDA neuronal subset specification. For instance, *Otx2* is expressed throughout the forebrain and midbrain at early stages of embryogenesis, whereas it is specifically expressed in the VTA of E18.5 embryos [92,93]. Interestingly, although expression of *Otx2* is known to be repressed by several factors during isthmus development (Figure 1A), it is currently unknown which factors are responsible for the specific expression of *Otx2* in the VTA subset. Specific deletion of *Otx2* by use of the *Dat*-cre driver does not result in a total loss of mdDA neurons in the VTA, indicating that *Otx2* is not required for the maintenance of these neurons, but is rather necessary for the induction and maintenance of a VTA identity [92]. However, postmitotic loss of *Otx2* results in an increase in the amount of dorsal-lateral *Girk2*-expressing neurons in the VTA, and a decrease in low-expressing *Dat* neurons, normally confined to the ventral lateral part of the VTA, suggesting that *Otx2* is involved in the development of different subsets of the VTA by promoting the expression of ventral-lateral VTA markers and inhibiting the expression of dorsal-lateral VTA markers [66,92]. These studies suggest a dual role for *Otx2* in mdDA neuronal development, one in FP patterning and expansion of the NPC pool, and one in subset differentiation of terminally differentiated mdDA neurons (Figure 2B).

Another gene with a dual role in DA specification and differentiation is *Lmx1a*, previously linked to proliferation and neurogenesis [48,51,53]. Besides its role in early midbrain specification, loss of *Lmx1a* has been shown to decrease the expression of *Vmat2* in both SNc and VTA areas, indicating that *Lmx1a* may function in specification of the entire mdDA neuronal population [94]. Furthermore, *Lmx1a* has been found to have a role in subset specification of the utmost rostrally located mdDA neurons [52]. *Lmx1a* is suggested to regulate the expression of *Nurr1* in the rostral mdDA neuronal population, the future SN [52]. In this domain, a loss of *Rspo2,* a regulator of canonical WNT-signaling, was detected [52]. Interestingly, *Rspo2* mutant mice show a partial phenocopy of the defect seen in the *Lmx1a* mutant, linking *Lmx1a* and subset specification to the regulation of canonical WNT-signaling [52] (Figure 2B). Furthermore, *Rspo2* has been suggested to promote mdDA neurogenesis in human embryonic stem cells and differentiation in both human embryonic stem cells and murine neuroblasts [95]. Taken together, these data do not only link *Lmx1a* to subset specification, but also suggests a role for WNT-signaling in the differentiation of mdDA subsets in both mouse and human mdDA neuronal development. Considering the amount of *Wnts* expressed through time in the embryonic midbrain and their disperse effects on mdDA neuronal differentiation, it is likely that regulation of the WNT-signaling cascade could play a major role in late subset specification of mdDA neurons, as was corroborated by recent studies. In a *Mest*/*Peg1* mutant, a protein that modulates WNT-signaling by inhibition of LRP6 maturation, a subset of the SN is lost, indicating that canonical WNT-signaling may be involved in the development of this specific group of cells [96] (Figure 2B). Furthermore, specific deletion of *Ezh2* at E12.5 via the *Pitx3-Cre* driver results in the affected maturation of mdDA neurons in the VTA and SN [97]. *Ezh2* has been shown to influence *Wnt1* expression, which could indirectly point to a role of *Wnt1* signaling in subset specification in the *Ezh2* mutant [31,97]. These data point to a role for WNT-signaling throughout mdDA neuronal development, from specification in the FP to differentiation into the different subsets.

Besides factors that function throughout mdDA neuronal development, specific factors are known that function only in mdDA neuronal differentiation. It has been shown that *Pitx3* is an important factor in the differentiation and maintenance of mdDA neurons [98,99]. *Pitx3* is expressed from E11.5 onward in TH-expressing neurons of the midbrain, and the loss of this factor results in the specific loss of AHD2-expressing neurons of the SN [98,99,100]. This specific loss of nigral neurons can be rescued by the addition of retinoic acid (RA) to *Pitx3* mutant embryos, although this is subset-specific, indicating that RA is an important factor in the regulation of nigral neuronal identity via *Pitx3* (Figure 2B) [100,101]. Although *Pitx3* is expressed in all TH-expressing mdDA neurons, loss of this factor severely affects the SN, whereas the VTA remains relatively intact [98,99]. This indicates that the function of *Pitx3* may differ between the subsets, or that VTA-specific genes can compensate for the loss of *Pitx3*. There are two factors in mdDA neuronal differentiation that have been shown to function side-by-side with *Pitx3*: *Nurr1* and *En1*. PITX3 was found to interact with NURR1 via binding to PSF [102]. This interaction is thought to release the repression of NURR1 by SMRT-HDAC complexes and results in the expression of NURR1 target genes upon recruitment of PITX3 [102]. This interplay of NURR1 and PITX3 is suggested to be subset-specific as the main effect on NURR1-regulated genes of PITX3 can be detected in the SN, indicating that in the VTA, this repression mechanism is either not present or regulated in a different way [4,102,103]. Besides an interaction with NURR1, it was recently found that *En1* and *Pitx3* show extensive and subset-specific cross-talk [104]. It was shown that loss of *Pitx3* results in an upregulation of *En1* and that these genes likely regulate each other’s expression [104]. Mutants of each of these genes show a loss of the rostro-lateral mdDA neuronal subset, indicating that they function in parallel to differentiate neurons of the future SN, whereas they show a reciprocal regulation of genes in the caudal mdDA neuronal subset, the future VTA, nicely illustrated by the loss of *Cck* in *En1* mutants and ectopic expression of this gene in *Pitx3* mutants [4,101,104]. Double *En1/Pitx3* mutants show a combined loss of *Ahd2* and *Cck* expression throughout the mdDA neuronal population [105]. However, a small subset of mdDA neurons still develops in these mutants, suggesting the existence of a mdDA neuronal subset independent of *Pitx3* and *En1* expression, which may be dependent on *Otx2* for their development [105]. Veenvliet and Smidt in their review of 2014 suggested a model for mdDA subset specification based on the interplay between *Pitx3*, *Nurr1*, and *En1*. They proposed that induction of *Nurr1* in DA progenitors generates a default mdDA neuron that can become part of the rostro-lateral subset or caudal subset. The definitive subset identity is then regulated by the differential interplay of *Pitx3* and *En1* [4].

Taken together, terminal differentiation and definitive subset identity are based on a tight interplay of genes that are expressed exclusively in mdDA neurons and genes that have a more broad expression and also function in the regulation of midbrain patterning, NPC proliferation, and early neurogenesis. Although not all factors that play a role in mdDA subset differentiation have been described here, the knowledge we have at the moment is insufficient to pinpoint what drives a DA progenitor to become an mdDA neuron and what makes an mdDA neuron part of a specific subset. Besides cell-intrinsic factors, extrinsic signaling cascades, like the different WNT-signaling pathways and SHH-signaling, may play a role in correct subset differentiation and should not be neglected in the search of cell-fate commitment and subset identification of mdDA neurons. Recently, several studies have been published that make use of single-cell RNA sequencing in mouse and human mdDA neurons during development and in the adult brain to gain more insight in the transcriptome of mdDA neuronal subsets [91,106,107,108]. These studies show that subset differentiation starts early in development, after which the subgroups segregate even further in molecularly distinct subsets both in the murine and the human brain [91,106]. These studies provide a detailed map of the transcriptome in single cells present in the mdDA neuronal population and validate earlier research on the molecular segregation of mdDA neuronal subsets [8,96,104]. However, some subsets of mdDA neurons are not detected in these studies, for instance, the *Rspo2*-expressing subgroup in the SN, affected in the *Lmx1a* mutant [52,91,106]. As only a limited amount of cells is sequenced, there is a significant loss of sequencing depth, resulting in a loss of information. This indicates that, although single-cell RNA sequencing can be used as a tool to determine novel factors expressed in mdDA neuronal subsets, some information about intrinsic and external factors involved in the development of these subsets may be lost. Therefore, a combination of single-cell RNA sequencing and more conventional genetics studies are important to obtain knowledge of the underlying mechanisms controlling subset development.

## 4. What Can We Learn from the Development of Other Systems in the Embryo?

Many factors that are necessary in the development of the embryonic midbrain are also known to play a crucial role in the development of other systems during embryogenesis. In order to further broaden our understanding of and gain more insight into the molecular mechanisms driving mdDA neuronal specification and differentiation, it is interesting to look into the development of other systems that make use of similar molecular mechanisms, like the immune system, cardiomyogenesis, and pancreatic cell development. Here, we will focus on immune system development in relation to development of the mdDA neuronal system, as the immune system contains different subsets of cells derived from one type of progenitor cells, similar to what is seen in mdDA neuronal development. Insight in subset development in the immune system may result in novel leads of different processes that are involved in mdDA neuronal subset development. Furthermore, the development of the mdDA neuronal population has many signaling pathways and transcription factors in common with the development of the immune system. Here, we will explain the interesting link between the development of the immune system and the mdDA neuronal (sub)population.

### Similar Factors Are Involved in Subset Development in the Immune System as in MdDA Neurons

As discussed above, the mdDA neuronal group of cells consists of different subsets, of which the SN and VTA are the most well-known due to the selective degeneration of the SN during PD [1]. Besides the mdDA neuronal population, more systems make use of different subsets within one large group of cells, like the adaptive immune system. It is known that the immune system makes use of at least three types of cells: The antibody-producing B-cells, the effectors of the cellular immune response T-cells, and natural killer (NK)-cells (Figure 3A) [109]. B-cells are generated in the bone marrow and arise from hematopoietic stem cells located in this structure, whereas T-cells arise from common lymphoid progenitor cells that migrate from the bone marrow to the thymus where they eventually form T-cells [109]. B-, T-, and NK-cells are all generated from the same hematopoietic stem cell lineage, and lineage commitment is achieved via the repression of lineage-inappropriate genes and activation of lineage-specific genes (Figure 3A), similar to what is seen during mdDA neuronal development when DA progenitors become mature mdDA neurons and acquire a specific identity [110,111,112,113,114].

Beside these three categories of immune cells, the T-cell lineage further specifies into different subsets of T-cells. T-cell progenitors specify into T-cells via several intermediate steps, or innate lymphoid cells (ILC2), a cell-fate that is actively repressed by E-box proteins expressed in T-cell progenitors, thereby favoring a T-cell cell-fate [113]. The first sub-division in T-cell formation is the division in the αβ cell lineage and the γδ cell-lineage from double-negative (DN) T-cell precursors that do not express CD4 or CD8 on their membrane (Figure 3A) [109,115]. αβ T-cells are most abundantly present in the population of circulating T-cells, whereas γδ T-cells are present in lower numbers and mainly in areas that are exposed to the exterior, like the gut [115]. γδ T-cells are part of the innate immune system and are generated upon strong signals from the T-cell receptor (TCR), whereas αβ T-cells are part of the adaptive immune system and are generated upon weaker signals from the TCR [115].

After division of αβ and γδ DN T-cells, αβ-expressing T-cells go through another round of subset specification. These T-cells become double-positive (DP), expressing both CD4 and CD8 on their cell membrane. This transition from DN to DP T-cells is thought to be under the control of SHH-signaling, WNT-signaling, and several bHLH proteins. SHH-signaling in the thymus has been found to negatively regulate differentiation from DN to DP T-cells, allowing for proliferation of DN cells and increasing the DN pool of T-cell progenitors [116,117]. Similar to SHH-signaling, WNT-signaling has been found to be involved in proliferation of DN T-cells to assure the size of the DN pool of T-cell progenitors (Figure 3A) [114,118,119,120]. DN T-cell precursors transit to the DP stage of T-cell precursors under the influence of a subfamily of the bHLH factors, namely the E-box proteins. E-box proteins are involved in B- and T-cell development, but, here, we will discuss their role in the specification of subsets in the T-cell population. It is thought that E-box proteins can compensate for each other’s function and that their role in T-cell development is highly dependent on dosage [121]. The main E-box protein involved in the DN-to-DP transition is *Tcf12*. Mice that are deficient in *Tcf12* have low numbers of DP T-cells, due to a stop in the transition from DN to DP T-cells (Figure 3A) [122,123].

When T-cells reach the DP state, they are further divided into different subsets. High levels of WNT-signaling at this stage will prime the DP-cell to specify into an invariant NK cell (iNK), which is part of both the innate and the adaptive immune system [112]. Lower or no levels of WNT-signaling favor the development of single-positive (SP) T-cells that express either CD4 that will become T-helper cells, or express CD8 cells that will become cytotoxic T-cells (Figure 3A) [124,125]. The ratio of CD4^+^ vs. CD8^+^ cells is 4:1, which is accomplished by lineage-specific cell death at key points during T-cell development [125]. SHH-signaling has also been found to be involved in the transition from DP to SP T-cells, as it was shown to favor the differentiation into CD8^+^ SP T-cells and negatively regulates CD4^+^ cell differentiation [124,126]. During the DP-to-SP stage transition of T-cells, the E-box protein *Tcf3* has mainly been implicated (Figure 3A). Loss of *Tcf3* leads to an accelerated transition from DP to SP T-cells, thereby depleting the pool of DP cells [123]. Proper regulation of E-box protein expression was shown to be critical to allow for the development of the CD4^+^ cell lineage, indicating that E-box proteins are important in the development of this subset of T-cells [127]. Interestingly, no specific function has been described for WNT-signaling in SP specification during T-cell development, although it has been shown that WNT3A regulates the maturation of T-cells in the periphery upon inflammation [114,128].

Above, we have shown that, similar to the mdDA neuronal population, cells of the immune system consist of different subsets. Transcription factor families and signaling cascades that have previously been found to be important in the regulation of mdDA neurogenesis are also involved in the specification of the different subsets of T-cells in the immune system. For instance, similar to midbrain development, SHH-signaling and WNT-signaling were detected in T-cell development to be importantly involved in extension of the progenitor pool [116,117] (Figure 3B). Furthermore, WNT-signaling was shown to be involved in expansion of the progenitor pool in T-cell development and the further fine-tuning of subset development, [114,128,129] similar to what has been described for mdDA neuronal development (Figure 3B). bHLH factors, and specifically E-box factors, appear to have an important role in development of the immune system at different levels during development [111,113,127]. E-box factors have recently been shown to be involved in mdDA neuronal specification and differentiation [63] (Figure 3B). To further elucidate their role in the development of this system, development of the immune system might hold important clues for their functioning in mdDA neuronal development. Therefore, it is likely that processes that act in T-cell differentiation could function in subset specification of the mdDA neuronal population. T-cell differentiation is not a one-step process and is characterized by the presence of several intermediate steps. This might similarly hold true for mdDA neuronal subset specification. All mdDA neurons are thought to arise from the FP in the midbrain VZ [9,12,37,43], indicating that they are specified by a relatively small set of transcription factors and signaling cascades. It is likely that during mdDA neuronal migration, mdDA neurons acquire their subset-specific cell-fate, which could also be marked by the presence of intermediate steps. Interestingly, T-cell lineage commitment is coordinated by signaling pathways and transcription factors that act at several steps of this process, indicating that the dosage of the different pathways and factors is of importance to acquire the correct lineage-fate. A similar process may be involved in mdDA neuronal subset differentiation, as both SHH and WNT-signaling have shown opposing results in the literature on their function during mdDA neurogenesis (Figure 3B).

Taken together, by examining similar developmental processes in different cellular systems during embryogenesis, we can gain more insight into mdDA neuronal subset development. We could take advantage of known developmental programs and culturing strategies successful for these systems and use them to develop culturing strategies to obtain the correct mdDA neuronal subset that can be used for cell replacement therapies in Parkinson’s disease.

## Figures and Tables

**Figure 1 ijms-21-04638-f001:**
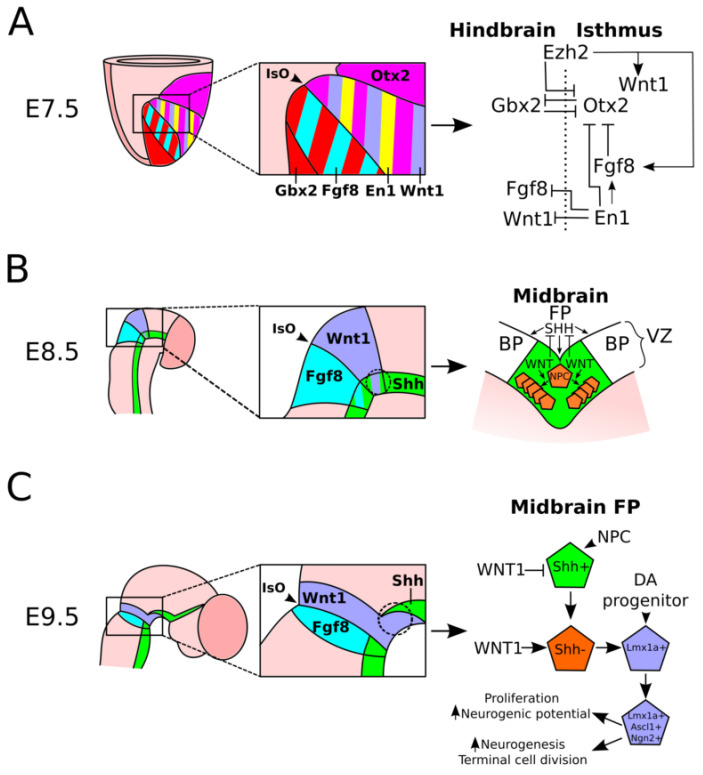
Patterning of the isthmus and midbrain region during early stages of development. (**A**): Schematic representation of the establishment of the isthmus during early stages of development. Around E7.5, the isthmus is established by the reciprocal expression of *Gbx2* and *Otx2*. This leads to expression of *Fgf8* in the isthmus, regulated by the expression of *En1* in the midbrain. *En1* inhibits the expression of *Wnt1* in the hindbrain, allowing for exclusive expression of this signaling molecule in the isthmus and midbrain area. The broadly expressed gene *Ezh2* positively regulates the expression of *Wnt1* and *Fgf8* in the isthmus and midbrain. (**B**): Schematic representation of the patterning of the FP by SHH and the increase in the neural progenitor cells (NPC) pool in the midbrain by WNT at early stages of development. Around E8.5, SHH-signaling in the FP of the midbrain allows for correct patterning of the FP and DA progenitors and WNT-signaling, inhibiting SHH signaling, for the expansion of the NPC pool in the VZ of the embryonic midbrain. (**C**): Schematic representation of the specification of DA progenitors from NPCs in the VZ of the mesodiencephalic FP. Expression of *Shh* in NPCs is inhibited by WNT-signaling in the midbrain area, which simultaneously activates the transition of NPCs to DA progenitors that express *Lmx1a*, *Ascl1*, and *Ngn2*. Expression of *Ascl1* in DA progenitors enhances proliferation and increases neurogenic potential, whereas *Ngn2* increases neurogenesis after terminal cell divisions.

**Figure 2 ijms-21-04638-f002:**
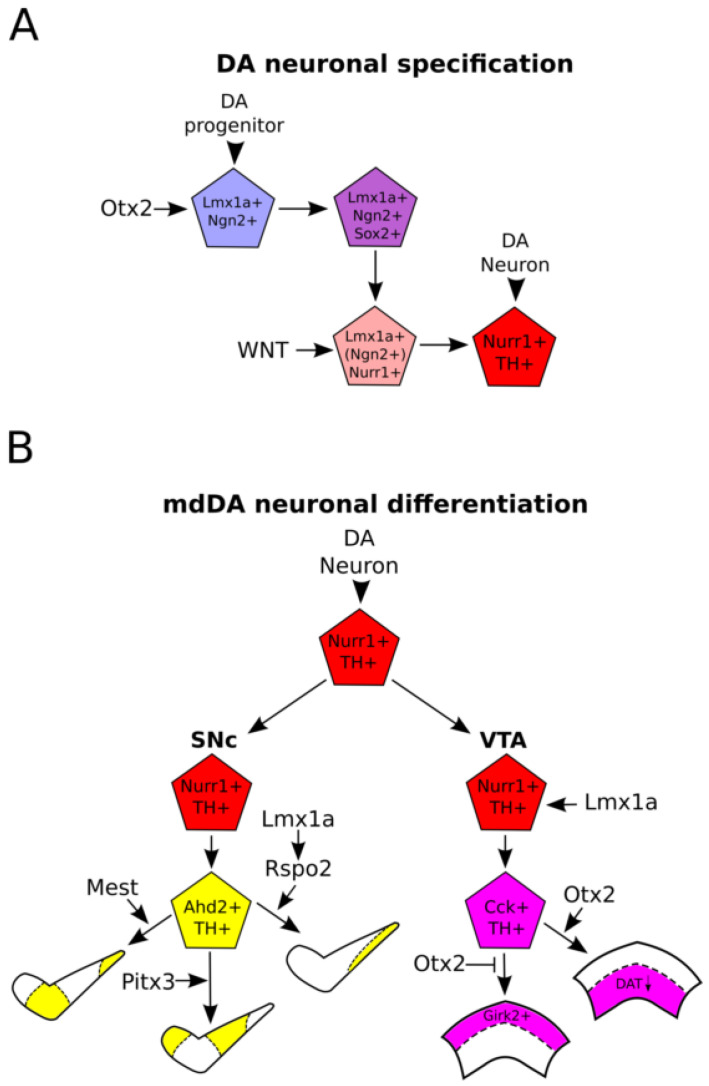
Specification of DA progenitors to a subset-specific mesodiencephalic dopaminergic (mdDA) neuron. (**A**): Schematic representation of the specification of a DA progenitor to an mdDA neuron. DA progenitors expressing *Lmx1a* and *Ngn2* (under control of *Otx2*) further progress in maturation after expression of *Sox2* in the VZ of the mesodiencephalic FP. *Lmx1a^+^/Ngn2^+^/Sox2^+^* DA progenitors leave the VZ and become *Lmx1a^+^/Nurr1^+^*, of which a small subset of intermediate precursors still express *Ngn2*. *Lmx1a^+^/Nurr1^+^* DA precursors become *Nurr1^+^/TH^+^* mdDA neurons under the influence of WNT signaling. (**B**): Schematic representation of subset differentiation of mdDA neurons. mdDA neurons that are *Nurr1^+^/TH^+^* can become part of the SN or VTA under the influence of different transcription factors and WNT-signaling. Some factors, like *Lmx1a*, are found to be involved in cell-fate commitment in both the SN and VTA, whereas other factors, like *Pitx3* and *Otx2*, are required for cell-fate commitment of the SN or VTA, respectively. MdDA neurons that become part of the SN typically start to express *Ahd2*, whereas neurons that become part of the VTA express *Cck*. After mdDA neurons are confined to a specific cell-fate, further subset specification is regulated by a unique combination of molecular factors. The WNT-regulating proteins MEST and RSPO2 (via LMX1A) are both involved in the development of a different subset in the SN, linking WNT-signaling to subset development. Interestingly, *Pitx3* appears to be involved in the development of another neuronal subset in the SN, which is not dependent on either *Mest* or *Lmx1a/Rspo2.* During VTA development, *Otx2* is involved in the development of the low-*Dat*-expressing neurons in the ventral-lateral VTA and inhibits the expression of *Girk2,* resulting in an *Otx2^-^Girk2^+^* neuronal dorsal-lateral population.

**Figure 3 ijms-21-04638-f003:**
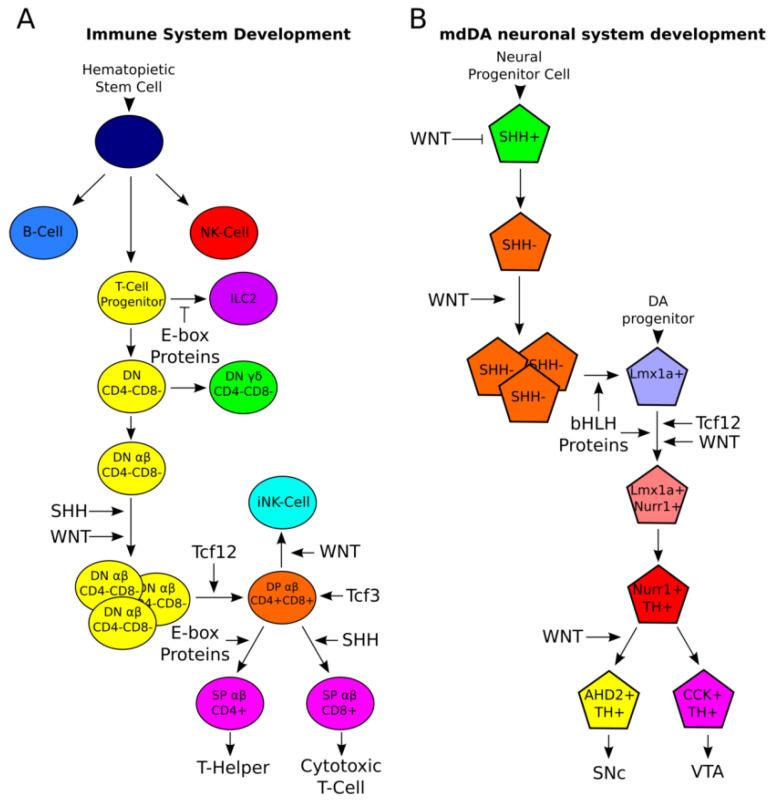
Development of the different subsets of the immune system. (**A**): Schematic representation of the development of the immune system. All different cell-types of the immune system arise from the same hematopoietic stem cell lineage. From these cells, the first three subsets in the immune system are derived: B-cells, T-cell progenitors, and NK-cells. T-cell progenitors can become innate lymphoid cells (ILC2), a fate repressed by E-box proteins expressed in T-cell progenitors, or further develop into double-negative (DN) γδ cells that become part of the innate immune system, and DN αβ cells that develop into CD4^+^CD8^+^ DP cells. High levels of WNT-signaling prime CD4^+^CD8^+^ DP cells to become innate natural killer cells (iNK), whereas E-box proteins and SHH signaling drive these cells into a CD4^+^ or CD8^+^ single-positive (SP) T-cell-fate, respectively. (**B**): Schematic representation of the development of the mdDA neuronal population. Similar to immune system development, all different cell types in the mdDA neuronal population arise from one group of neural progenitors. WNT- and SHH-signaling are known to be involved in expansion of the progenitor cell pool and further specification into DA neurons. Similar to immune system development, later fine-tuning into the different molecular subsets is regulated by bHLH and E-proteins, like *Tcf12*, but also by signaling cascades like WNT. Considering the similarities between the development of these systems, molecular mechanisms involved in the development of the immune system may hold clues for the development of the mdDA neuronal population and vice versa.

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
