# Peer review of "Acquisition of the Midbrain Dopaminergic Neuronal Identity"

_ijms, 2020, doi:10.3390/ijms21134638_

Round 1
Reviewer 1 Report
In their Review article, Mesman and Smidt summarize more recent and older findings regarding the initial patterning of the midbrain region leading to the establishment of the midbrain dopaminergic progenitor domain in the midbrain floor plate, and the subsequent specification of the midbrain dopaminergic cell fate in these progenitors and their differentiation into mature midbrain dopaminergic neurons, especially of the substantia nigra pars compacta (SNc) and ventral tegmental area (VTA) subsets. At the end, the authors propose that midbrain dopaminergic neuron development shares several regulatory features of genes and signaling pathways with the development of the immune system. They also propose that comparison of both developmental systems may provide yet unknown clues about the precise steps in midbrain dopaminergic neuron development. Neurons making the neurotransmitter dopamine in the mammalian midbrain remain in the focus of biomedical research because of their involvement in several neurological and psychiatric human disorders, including Parkinson’s Disease.
Regarding its content, the Review by Mesman and Smidt adds a newer perspective to the already vast literature on midbrain dopaminergic neuron development. However, some aspects of their Review article could be more elaborated, and the underlying literature is improperly cited or not cited at all in some instances. The following improvements of their article are therefore suggested:
1) This Review article deals with the initial patterning of the ventral midbrain dopaminergic progenitor domain, the subsequent specification of the dopaminergic fate in neural precursor cells, and their differentiation into the different midbrain dopaminergic neuron subsets, but anything at all with the maintenance of their neuronal identity in the adult mouse brain. The title of this article is thus somewhat misleading and could be better adapted to its content.
2) The animal species, to which most of its content about midbrain dopaminergic neuron development is referring to, could be clarified at the beginning of this Review article.
3) Figure 1 and related text on pages 2 – 5:
- The schematic representations of the embryonic mouse (?) brain in Fig. 1A and 1B are somewhat inauspicious. Firstly, at these early embryonic stages (E7.5 and E8.5), the neural tube has not yet formed and the “brain” consists of a neural plate that starts to fold up at E8.5. Secondly, the expression domains of Wnt1, Otx2, Gbx2 and En1, in particular, are much broader at these developmental stages than depicted in Fig. 1A. The apparently ventral restriction of Otx2, Gbx2 and En1 expression in this figure suggests that these factors are expressed in a similar manner as Shh in the floor plate at E8.5, and is thus totally misleading. Gbx2 and En1 expression, on the other hand, do not extend into the caudal hindbrain. This figure should thus be corrected.
- Is WNT1 signaling really active in the midbrain floor plate before E11.5, as suggested in this figure and in the accompanying text? The authors could take into consideration publications from the Arenas and Wurst labs showing that potent inhibitors of this signaling pathway, namely of the Dickkopf (Dkk) family, are expressed in this domain up to this developmental stage.
- Accordingly, the dosage of WNT signaling might be another explanation for the improper specification of midbrain dopaminergic neurons after ectopic expression of (active) b-catenin, apart from patterning and timing that are referred to in lines 137 – 138.
- Redundancy between the orthologue genes Lmx1a and Lmx1b could be another explanation for the subtle phenotypes in the Lmx1a single mutants described in the second last and last paragraph of section 2.2. (“FP patterning in the midbrain”), and the corresponding literature could be included here as well.
- Timing of bHLH transcription factor expression (and function) in the midbrain floor plate could also be an explanation for the different midbrain dopaminergic phenotypes of the corresponding mouse mutants, and could be discussed in the last paragraph of this section (2.2.) as well.
2) Figure 2 and related text on pages 5 – 10:
Literature in this part of the article is sometimes improperly cited or missing at all. For example:
- The depletion of caudal midbrain dopaminergic neuronal pools in conditional Otx2 mutants (lines 174 – 178) was demonstrated more conclusively in later publications from the Simeone lab, e.g. in Omodei et al. 2008 and DiGiovannantonio et al. 2014.
- The reference to Metzakopian et al. 2015 (reference # 65) in line 184 seems to be out of context in this paragraph. This reference, however, could well be included in the third paragraph of section 3.2. (“Late differentiation and mdDA subset specification”). This paragraph deals with epigenetic factors involved in midbrain dopaminergic subset specification, and Smarca1/Snf2l is another of these factors implicated in midbrain dopaminergic neuron development.
- The proof of an influence of WNT signaling in the specification of midbrain dopaminergic neuron subsets is still weak at present, and the authors should clarify this point. Again, dosage effects of WNT signaling should also be taken into consideration, and the corresponding literature could be cited here as well.
- What are “post-mitotic NURR1-expressing DA progenitors” mentioned in line 210 – 211?
- The authors should clarify which WNTs they are referring to in line 227?
- The reference to the original paper (Castelo-Branco et al., 2003) implicating WNT3A in mdDA neuron differentiation (is that really true?) is missing in line 242. Apart from miRNAs downregulating Wnt1 levels in mdDA progenitors, attenuation or inhibition of canonical WNT1 signaling in these cells could also be considered as a mechanism enabling the differentiation of proliferating mdDA progenitors into mdDA neurons (lines 245 – 248), and the corresponding literature could be cited here as well.
- What is the evidence that “most mdDA neurons arise from one small part in the developing midbrain, the (caudal) FP” (lines 267 – 268)? The cited literature in line 268 not necessarily demonstrates this.
- The text in lines 289 – 291 is misleading. Di Salvio et al. (2010) have only shown that deletion of Otx2 in postmitotic midbrain dopaminergic neurons results in the loss of certain marker proteins for the ventrolateral VTA and expansion of dorsolateral VTA markers, and this correlates with the survival capabilities of these neurons under adverse conditions. However, these data cannot be interpreted as if “Otx2 acts as an inhibitor in the development of the neuronal population of the dorsal-lateral VTA and an activator in the development of the neuronal population of the ventral-lateral VTA”, as stated in the text.
- Apart from their own work, other published evidence (e.g. Gyllborg et al. Stem Cell Rep., 2018) regarding the expression and function of the regulator of canonical WNT-signaling, Rspo2, in midbrain dopaminergic progenitors/neurons (lines 299 – 302 and 357 – 359) could be cited and discussed by the authors.
3) Figure 3 and related text on pages 10 – 13:
- Although the insights that might be obtained from establishing analogies between midbrain dopaminergic neuron development and the development of the immune system are appreciated, the authors should point out the specific lessons we might learn from the immune system and clarify that this may be one of several other possible examples. In fact, similar developmental programs as in the midbrain dopaminergic system are also found during cardiomyogenesis and in pancreatic cell development, indicating that these analogies might not be exclusive for the immune system. So what can we learn specifically from the immune system, and how can this bring us forward in the design of proper cell replacement strategies for Parkinson’s Disease?
Apart from these suggestions, there are some other minor issues as specified below:
1) The abbreviations “mdDA” (for mesodiencephalic dopaminergic) and “DA” (for dopamine, but probably meaning also dopaminergic) are inconsistently used throughout the text and should be unified.
2) Several abbreviations used in the text, as well as abbreviations for e.g. growth factor names, are missing in the list.
3) Some editing of the text is needed, as several mistakes are still found in it (e.g., extension OF, reciprocal instead of reciproke, hematopoietic instead of hemopoietic, Basal plate instead of Asal plate in the abbreviations).
Reviewer 2 Report
The authors summarized several neuronal factors that characterize the subsets of dopaminergic neurons. Specifically, they provided the detailed examples how those several representative factors are involved in the development process of the neurons. This is a very interesting and well summarized review paper. The present manuscript is of sufficient impact and general interest to warrant publications in International Journal of Molecular Sciences.
In this review article, the authors well described the identity of dopaminergic neurons comprehensively.
1. Generally SNc stands for substantia nigra pars compacts, a division of substantia nigra (SN). Please use a universal abbreviation and clarify if the authors described all the characters of SNc or SN.
Round 2
Reviewer 1 Report
The authors have satisfactorily addressed all previous concerns of this Reviewer.